# Optimization and Development of Selective Histone Deacetylase Inhibitor (MPT0B291)-Loaded Albumin Nanoparticles for Anticancer Therapy

**DOI:** 10.3390/pharmaceutics13101728

**Published:** 2021-10-19

**Authors:** Athika Darumas Putri, Pai-Shan Chen, Yu-Lin Su, Jia-Pei Lin, Jing-Ping Liou, Chien-Ming Hsieh

**Affiliations:** 1School of Pharmacy, College of Pharmacy, Taipei Medical University, Taipei 11031, Taiwan; d301108007@tmu.edu.tw (A.D.P.); m301107002@tmu.edu.tw (Y.-L.S.); linjpjpjp@gmail.com (J.-P.L.); jpl@tmu.edu.tw (J.-P.L.); 2Graduate Institute of Toxicology, College of Medicine, National Taiwan University, Taipei 10617, Taiwan; paishanchen@ntu.edu.tw

**Keywords:** albumin nanoparticle, MPT0B291, high-pressure homogenizer, histone deacetylase

## Abstract

Histone deacetylase (HDAC) inhibitors have emerged as a new class of antitumor agent for various types of tumors. MPT0B291, a novel selective inhibitor of HDAC6, demonstrated significant antiproliferative activity in various human cancer cell types. However, MPT0B291 has very low water solubility, which limits its clinical use for cancer therapy. In the current study, MPT0B291 was encapsulated in human serum albumin (HSA), and its anticancer activities were investigated. Nanoparticles (NPs) were prepared using two-stage emulsification resulting in 100~200-nm NPs with a fine size distribution (polydispersity index of <0.3). The in vitro drug release profiles of MPT0B291-loaded HSA NPs presented sustained-release properties. The cytotoxic effect on MIA PaCa-2 human pancreatic carcinoma cells was found to be similar to MPT0B291-loaded HSA NPs and the free-drug group. The albumin-based formulation provided a higher maximum tolerated dose than that of a drug solution with reduced toxicity toward normal cells. Furthermore, in vivo pharmacokinetic studies demonstrated an effective increase (5~8-fold) in the bioavailability of NPs containing MPT0B291 loaded in HSA compared to the free-drug solution with an extended circulation time (*t*_1/2_) leading to significantly enhanced efficacy of anticancer treatment.

## 1. Introduction

Histone acetyltransferases (HATs) and histone deacetylases (HDACs) are known to simultaneously regulate both intracellular and extracellular responses toward epigenetic modifications by respectively executing the acetylation and deacetylation of lysine residues at the amino terminals of histones [1,2]. However, overexpression and atypical recruitment of HDACs for acetyl-group removal that occurs at their promoter sites raises the possibility of an imbalance in acetylation-deacetylation processes [3,4,5]. Aberrant deacetylation due to the HDAC mechanism was also found in non-histone proteins, such as p53, which initiated a mutation of p53 [6]. Consequently, targeting HDACs is considered an important strategy in developing anticancer agents.

Inhibition of HDACs has garnered potential interest in anticancer development as it induces cell-cycle arrest and cell apoptosis, and decreases cancer metastasis [7]. Several recently developed HDAC inhibitors, such as deacetylase-trichostatin A (TSA) and deacetylase-suberoylanilide hydroxamic acid (SAHA), showed excellent results in suppressing cancer cell growth [8,9,10].

Among various mammalian HDAC isozymes, HDAC6 is unique for being restricted to the cytoplasm and possessing two catalytic domains along with a ubiquitin-binding site [11]. One of the domains, CD2, induces deacetylation of the α-tubulin region of microtubules. Therefore, targeting HDAC6 could be a prospective cancer immunotherapy, as inhibition of its site showed improvement in tumorigenesis, through reducing histone hypoacetylation and increasing tumor-suppressor genes [12,13,14].

MPT0B291, an azaindolysulfonamide (shown in Figure 1), was developed as an HDAC6 inhibitor and is currently being studied for applications in clinical treatments. Compared to SAHA, MPT0B291 is considered a promising anticancer drug candidate due to its higher selectivity of HDAC inhibition, low toxicity, and broader spectrum of anticancer activity [14,15]. Despite this significant therapeutic potential, MPT0B291 has very low water solubility, which directly impacts its pharmacokinetics (PKs) and ultimately limits its clinical use. In addition, the poor aqueous solubility of MPT0B291 hinders its formulation for parenteral delivery.

Utilization of nanoparticles (NPs) for drug delivery and therapy has been developing for decades. NPs’ fundamental characteristic of a high surface-to-volume ratio permits interaction with a greater surface area in comparison to that of larger particles with a similar volume [16]. This unique advantage in drug development enhances the drug-loading capacity and stability, and in a more feasible way enables the incorporation of ligands (i.e., either hydrophilic or hydrophobic molecules). Protein-based carriers have been extensively applied owing to their potential advantages over synthetic carriers, including low toxicity, high drug-binding capacity, marked uptake by target cells, easy preparation, and scaling-up capability [17,18].

Among successful drug nanocarriers that are now established, albumin-bound NPs have opened up an avenue as an effective nanocarrier in cancer therapeutics [19]. Albumin can closely and reversibly bind to hydrophobic paclitaxel through noncovalent bonds to enable in vivo transport and release of transported substances and has become a natural carrier of hydrophobic substances [20,21,22]. It can also provide improved efficacy and tolerability compared to cremophor-based paclitaxel solutions [23,24]. Using albumin as a nanocarrier is essential to disguise the active loaded drugs from unselective uptake by immune cells. The use of the first albumin-bound NPs to facilitate delivery of the anticancer drug, paclitaxel, has garnered tremendous interest, as they are able to increase the stability of the colloidal solution (Abraxane^®^). Three essential pathways, known as ‘tumor-feeding mechanisms’, corroborate the importance of albumin as a drug nanocarrier. High permeability of tumor neo-vessels permits more macronutrients, including albumin, to leak inside the tumor area than that seen with normal tissues, implying indirect contact of the co-loaded cytotoxic species which use albumin as the template core. Second, albumin trapped within a tumor is slowly digested due to the impaired lymphatic drainage which can lead to efficient targeting or effective binding of the co-loaded species with the targeted receptor. Third, the albumin-activated gp60 route allows albumin transcytosis to bind to the gp60 receptor on vascular endothelial cells [25].

Therefore, in the present study, we aimed to develop, characterize, and optimize novel nanocarrier formulations of MPT0B291 with the relatively inexpensive, commercially available, safe, and biocompatible human serum albumin (HSA). MPT0B291-loaded HSA NPs (MPT0B291-HSA NPs) were prepared, and their size distribution, morphology, and loading and encapsulation efficiencies were examined. The in vitro cytotoxicity of the optimized formulation was assessed against the Mia PACA-2 cell line. Finally, in vivo PK studies were performed to investigate whether albumin encapsulation could prolong the elimination half-life and improve the intravenous bioavailability of MPT0B291.

## 2. Materials and Methods

MPT0B291 was synthesized in our own laboratory (Taipei, Taiwan) [15]. BSF (20%) human albumin solution (≥96.0%, CSL Behring, Melbourne, Australia), soybean lecithin (Lipoid S100) (Lipoid, Ludwigshafen am Rhein, Germany), Bio-Rad protein assay dye reagent concentrate (Bio-Rad Laboratories, Hercules, CA, USA), bovine serum albumin, (BSA; Sigma-Aldrich Chemie, Steinheim, Germany), methanol, ethanol, acetonitrile, chloroform, Tween 80 (Merck, Darmstadt, Germany), dimethyl sulfoxide (Echo, Miaoli, Taiwan), formic acid, (Sigma-Aldrich Chemie), and trifluoroacetic acid (Riedel-de Haėn, RdH Laborchemikalien, Seelze, Germany) were procured commercially. All organic solvents mentioned above were of analytical grade. Sodium phosphate monobasic (Sigma-Aldrich Laborchemikalien, Seelze, Germany), sodium phosphate dibasic, anhydrous (J.T. Baker, Phillipsburg, NJ, USA), Dulbecco’s modified Eagle medium (DMEM), fetal bovine serum (FBS), horse serum, penicillin-streptomycin solution, 0.25% Trypsin-2.21 mM EDTA 1×, Matrigel^®^ matrix basement membrane (Corning, Manassas, VA, USA), and thiazolyl blue tetrazolium bromide (Alfa Aesar, Lancashire, UK) were purchased commercially. Male Sprague-Dawley rats at 8 weeks of age and female fox chase SCID mice at 4 weeks of age were purchased from BioLASCO Taiwan (Yilan, Taiwan). All other chemicals and solvents used in this study were of fine analytical grade.

### 2.1. Preparation of MPT0B291-Loaded Albumin NPs

MPT0B291-HSA NPs were prepared by an emulsion-solvent evaporation method using a high-pressure homogenizer. Briefly, mixtures of MPT0B291 (30 mg) and soybean lecithin (135 mg) were dissolved in 800 µL of chloroform/ethanol (9:1). The solution was heated in an ultrasonic bath to 50 °C until the drug had completely dissolved. The drug-containing solution was then added to the albumin aqueous solution (1% *v*/*v*) with 5 min of sonication under 30% or 50% amplitude (ultrasonic oscillation; VCX 750, frequency: 20 kHz, Sonics and Materials, Newtown, CT, USA). The first emulsion was obtained followed by high-pressure homogenization (NanoLyzer-N2, Cogene, Hsinchu, Taiwan). A homogenization pressure (15,000 and 20,000 psi) was applied to the emulsion, and the number of homogenization cycles (10, 20, 30, and 40 cycles) was also optimized. The emulsion was subjected to various homogenization cycles, then passed through the homogenizer valve, and collected through a connecting tube at the base of the assembly, thus forming nano-sized emulsion droplets. Following high-pressure homogenization, the resulting solution was transferred to a round-bottomed flask and subjected to rotary evaporation (R-114, Buchi, Switzerland) under low vacuum (500 mmHg) for 25 min at 50 °C in order to ensure that the organic solvent was completely removed. The evaporated NPs were then filtered with 0.2 µm-regenerated cellulose (RC) membrane filter to remove impurities and unbound drugs. Finally, the NPs were freeze-dried (FDS-2, Taipei, Taiwan) using trehalose as cryoprotectant and stored at 4 °C for further experiments. A 3% trehalose solution was employed to stabilize the HSA NPs as well as to prevent the protein from denaturation during the lyophilization process and provide protection for long-term storage [26,27].

### 2.2. Particle Size, Size Distribution, Zeta Potential, and Surface Morphology

The particle size, size distribution, and zeta potential of the HSA NPs and MPT0B291-HSA NPs were measured by dynamic light scattering (DLS) using a Malvern Zetasizer NanoZS (Malvern Instruments, Worcestershire, UK). Samples were diluted with deionized water and measured at a scattering angle of 90° and temperature of 25 °C. The polydispersity index (PDI) gave an estimate of the size distribution of the HSA-NPs. The zeta potential was measured with a zeta potential analyzer using electrophoretic laser Doppler anemometry (Malvern Instruments, Worcestershire, UK). In addition, the shape and surface morphology of the MPT0B291-HAS NPs were further examined by transmission electron microscopy (TEM) (HT7700, Hitachi, Tokyo, Japan). Samples were diluted 50× with distilled water, and the NPs were stained with the heavy metal salt, uranyl acetate (2 *w*/*w*%), prior to imaging. A sample was prepared by placing a drop of the HSA NPs on a carbon-coated copper grid under suitable conditions.

### 2.3. Encapsulation Efficiency (EE) and Drug Loading (DL) of MPT0B291-HSA NPs

The drug concentration was initially determined by high-performance liquid chromatography (HPLC) using a Shimadzu-20AD system (Kyoto, Japan). The HPLC system was equipped with an XBridge C18 column (particle size 5 µm, 4.6 × 150 mm). A mixture of 0.043 M ammonium acetate and acetonitrile (45:55, *v*/*v*) was used as the mobile phase at a flow rate of 1 mL/min at 40 °C. The column effluent was monitored using an ultraviolet detector (UV-975, Jasco, Tokyo, Japan) at a wavelength of 265 nm, and the HPLC method was validated to have an acceptable coefficient of variation for accuracy and precision, which met the criteria of ±15% [28].

The drug content of MPT0B291-HSA NPs was determined via a literature search, which is briefly summarized as follows [29]. To release the drug content from the NPs, the MPT0B291-HSA NPs were diluted in acetonitrile and sonicated for 15 min. The supernatant obtained after centrifugation at 10,000 rpm for 5 min was then injected into the HPLC system to quantify the amount of MPT0B291 at 265 nm. The equations used for calculating the DL (%) and EE (%) are as follows:(1)$$\mathrm{Encapsulation}\;\mathrm{efficiency}\;\left(\mathrm{EE},\;\%\right)=\frac{W_{\left(\mathrm{Total}\;\mathrm{drug}\;\mathrm{added}-\mathrm{free}\;\mathrm{non}-\mathrm{encapsulated}\;\mathrm{drug}\right)}}{W_{\mathrm{Total}\;\mathrm{drug}\;\mathrm{added}}}\times 100\%$$
and
(2)$$\mathrm{Drug}\;\mathrm{loading}\;\left(\mathrm{DL},\;\%\right)=\frac{W_{\left(\mathrm{initial}\;\mathrm{drug}\;\mathrm{added}-\mathrm{free}\;\mathrm{non}-\mathrm{encapsulated}\;\mathrm{drug}\right)}}{W_{\mathrm{nanoparticles}\;\mathrm{content}}}\times 100\%$$

### 2.4. Stability of MPT0B291-HSA NPs

The MPT0B291-HSA NPs were reconstituted in water, and their sizes were measured by DLS at regular time intervals. In addition, the total drug concentrations were evaluated after 7 days of storage at 4 °C.

### 2.5. In Vitro Drug-Release Behavior

To investigate the kinetics of MPT0B291 release from MPT0B291-HSA NPs, a suspension of MPT0B291-HSA NPs containing 0.25 mg/mL of MPT0B291 was transferred to a dialysis bag with a molecular weight cutoff (MWCO) of 6000~8000 Da. The dialysis bag was subsequently placed into 25 mL of release medium (PBS) containing 1% *v*/*v* Tween 80. The temperature was maintained at 37.0 ± 0.5 °C, and the medium was stirred at a speed of 100 rpm. MPT0B291 which had been dissolved in propylene glycol (free MPT0B291; 0.25 mg/mL) was used as a control and was treated in the same manner. During a period of 48 h, the surrounding environment was maintained by replacing 2 mL of the release medium with an equal volume of fresh medium at regular intervals. The extraction solution was solubilized in 500 μL of acetonitrile and assayed by HPLC.

### 2.6. Cell Viability Assay

The cytotoxicity of the MPT0B291-HSA NPs was determined in MIA PaCa-2 cells using MTT assays. Briefly, MIA PaCa-2 cells were seeded (5 × 10^3^ cells/well) in 96-well culture plates. After 24 h of incubation in a humidified incubator at 37 °C with 5% CO_2_, the medium of adherent cells was replaced with serum-free culture medium and incubated with 200 μL of serial dilutions of HSA-NPs (0.01~1000 μg/mL) and MPT0B291-HSAX NPs (0.0050 μg/mL). After another 24 h, 100 µL of MTT (500 μg/mL in PBS, pH 7.4) was added and incubated at 37 °C for 2 h. The medium was then removed, and 100 µL of DMSO was added to each well to dissolve the crystals. The optical density (OD) was measured at 540 nm using a microplate reader (Cytation™ 3 Cell Imaging Multi-Mode Reader, BioTek, Winooski, VT, USA). The cell survival rate was calculated as follows:(3)$$\mathrm{Cell}\;\mathrm{survival}\;\mathrm{rate}\;\left(\%\right)=\frac{\mathrm{Absorbance}\;\mathrm{value}\;\mathrm{of}\;\mathrm{test}\;\mathrm{solution}\;\mathrm{group}\;}{\mathrm{Absorbance}\;\mathrm{value}\;\mathrm{of}\;\mathrm{blank}\;\mathrm{solution}\;\mathrm{group}\;}\times 100\%$$

### 2.7. Maximum Tolerated Dose (MTD)

The MTD studies for the MPT0B291 solution and MPT0B291-HSA NPs were carried out in healthy Balb/c female mice. The single-dose study was conducted using healthy female mice with four animals per group. Two groups of mice received an i.v. injection of 50, 75, or 150 mg/kg body weight (BW) of MPT0B291 and 0, 75, 150, or 200 mg/kg BW of MPT0B291-HSA NPs through the tail vein. The MTD was defined as the allowed median BW loss of approximately 15% of the control and causing neither death due to toxic effects nor remarkable changes in general signs within 1 week of administration [30].

### 2.8. In Vivo PK Study

Male Sprague-Dawley rats (250 ± 15 g) were divided into two groups (with four rats per group). Rats were fasted for 12 h with free access to water, and then intravenously treated with the MPT0B291 solution (dissolved in propylene glycol) or MPT0B291-HSA NPs (relative to the MTP0B219 concentration of 5 mg/kg BW). Blood samples (approximately 0.5 mL) were collected in a heparinized tube at 5 min, 15 min, 30 min, 1 h, 2 h, 4 h, 6 h, 8 h, 12 h, and 24 h after drug administration. All collected blood samples were centrifuged at 4000 rpm and 4 °C for 10 min, and supernatant plasma samples were obtained and frozen at −80 °C for further analysis.

Quantitative analysis of MPT0B291 in rat plasma by an ultra-precision liquid chromatographic tandem mass spectroscopic (UPLC-MS/MS) method was performed using an Agilent 1290 Infinity II LC System (Agilent, Wilmington, DE, USA) coupled with an Agilent 6470 triple quadrupole LC/MS system (Agilent Technologies, Santa Clara, CA, USA). Optimized chromatographic separation of MPT0B291 was conducted with an ACQUITY UPLC BEH C18 column (1.7 µm, 2.1 × 100 mm; Waters, Milford, MA, USA) at an oven temperature of 40 °C. The mobile phase was 0.1% (*v*/*v*) formic acid in water (mobile phase A) and acetonitrile containing 0.1% (*v*/*v*) formic acid (mobile phase B) with gradient elution at a flow rate of 0.3 mL/min. All analytical procedures were evaluated with positive electrospray ionization (ESI). The LC gradient was programmed in linear steps as follows: 0~0.5 min, 5% mobile phase B (*v*/*v*); 0.5~1 min, 5% mobile phase B to 100% mobile phase B (*v*/*v*); 1~2.8 min, 100% mobile phase B; 2.8~3 min, 100% mobile phase B to 5% mobile phase B (*v*/*v*); and 3~4 min, 5% mobile phase B (*v*/*v*). Quantification was achieved using the multiple reaction monitoring (MRM) mode at *m*/*z* 344.1 →  91.0 for MPT0B291. The following PK parameters were determined: the area under the concentration-time curve (AUC), maximum peak concentration (Cmax), time of maximum peak concentration (Tmax), and mean residence time (MRT). These parameters were determined by DAS software vers. 2.0. All data are presented as mean ± standard deviation (SD).

### 2.9. Statistical Analysis

All data were sourced from multiple batches and expressed as the mean ± SD unless specifically stated. Student’s *t*-test or a one-way analysis of variance (ANOVA) was performed for the statistical evaluation of data. Differences between groups were considered statistically significant when the probability (*p*) was <0.01.

## 3. Results and Discussion

### 3.1. Optimization and Preparation of HSA NPs

#### 3.1.1. Effect of the Energy Output of Ultrasonication

The fact that there is an enhanced permeability retention (EPR) effect around the tumor area is well-established, a higher accumulation rate of blood-circulating proteins including albumin would also be predicted [31]. Therefore, limiting the NPs to a sufficiently small size, within or <250 nm along with a uniform particle distribution (with an expected polydispersity index (PDI) of <0.30), to facilitate prolonged NP circulation and penetration into tumor tissues via an EPR effect is critical. It is also important to note that using a higher albumin concentration for preparation of albumin NPs would be anticipated to initiate higher levels of disulfide bonds formation leading to increased particle size and a greater likelihood of protein aggregation [32,33,34]. Further, higher particle size might affect the lower volume-to-surface ratio of the NPs, in turn decreasing drug loading during drug-NPs preparation and cellular uptake in vivo [35,36,37,38]. Lomis and co-worker [33] observed that 1% of albumin would be appropriate as the starting concentration of HSA NPs processed through HPH [33]. Therefore, 1% albumin was selected in this study for the preparation of HSA NPs and MPT0B291-HSA NPs. The HSA NPs were prepared by a two-step emulsification method (sonication and homogenization). Optimizing HSA NPs’ preparation is a complex process due to the wide range of parameters involved in controlling their size. In this study, we investigated the influences of the sonication amplitude and homogenization process on the average diameter of HSA NPs. As shown in Figure 2, two different energy inputs (30% and 50% sonication amplitudes), followed by various homogenization cycle numbers (10, 20, 30, and 40), under 20,000 psi of homogenization pressure, were investigated to understand their effects on the size distribution of HSA NPs. An improved dispersion was found with a 30% amplitude of sonication regardless of the homogenization cycles, while a slightly larger particle size and wider particle size distribution were found when using a 50% sonication amplitude.

In general, a higher ultrasonication amplitude would be expected to create a higher frequency of collisions between droplets and to further decrease the duration required to achieve a minimum emulsion droplet size. Fragmentation of the dispersed phase by ultrasonic radiation is mainly caused by a cavitation effect generating shear stress, local high pressure, and increased temperature [39]. A higher energy output can generate a more significant cavitation effect [40] which ultimately reduces the diameter of NPs. However, the activity of a hydrophobic drug counterpart involved in non-covalently bonding formation may have been interrupted by use of higher energy during the ultrasonication process. In the current study, applying a high amount of energy may have resulted in further instability of the formulation leading to particle aggregation. Ineffective promotion of a uniform distribution of particle sizes was found in groups formed with a 50% amplitude. It can be concluded that using a high sonication amplitude appeared to have a negative impact on the particle sizes obtained by the two-step emulsification method.

#### 3.1.2. Effect of the Number of Homogenization Cycles and Pressure

High-pressure homogenization (HPH) is widely acknowledged to promote stable cross-linking of albumin through disulfide bonds in the binding of albumin NPs to paclitaxel (i.e., nab-Paclitaxel) [41]. In this study, HPH was utilized to enhance process emulsification in the second stage to facilitate smaller particle sizes and generate a more-homogeneous size distribution of NPs. With a fixed MPT0B291/HSA ratio following ultrasonication with the amplitude set to 30% for 5 min, the influence of homogenization pressures (15,000 and 20,000 psi) and various cycle numbers (10~40 cycles) were evaluated to study their impacts on the stability of NP formation and the size distribution at a fixed MPT0B291/HSA ratio. In Figure 3, the particle size increased with an increase in HPH cycles from 0 to 40 cycles when 15,000 psi was applied as the homogenization pressure. When utilizing 20,000 psi of homogenization pressure, the particle size increased between 10 and 30 cycles but then dramatically decreased to less than 100 nm. The smallest particle diameter (75.8 ± 0.86 nm) was obtained at a homogenization pressure of 20,000 psi with 40 cycles. Mohan and Narsimhan found that a higher turbulent intensity could lead to higher coalescence efficiencies and higher rates of collision between drops due to the larger turbulent forces, thus resulting in a higher coalescence rate [42]. In this case, a higher pressure of 20,000 psi seemed to affect the efficiency of the emulsifying properties leading to a higher coalescence rate after the homogenizing valve.

The PDI decreased with an increase in homogenization cycles from 10 to 30 regardless of the homogenization pressure and reached the lowest point at 30 cycles with homogenization pressures of both 15,000 and 20,000 psi. However, when the number of cycles increased to 40, the PDI inversely increased for both homogenization pressures. The increased size observed with an increasing number of cycles may have been caused by the coalescence of very small droplets passing through the HPH valve at elevated pressures. Our results indicated that the homogenization pressure did not markedly influence the droplet size distribution, whereas the number of homogenization cycles was more influential on the particle size and size distribution in this study. Increasing the number of homogenization cycles can improve the homogeneity of the population by eliminating the few remaining large particles and reducing the PDI [43]. Although the pressure did not have a direct effect on the NP size when the NPs were prepared by HPH, increasing the number of homogenization cycles combined with the applied pressure led to a greater reduction in particle size, as shown in the case of 20,000 psi and 40 cycles. Several important factors influence the quality of NPs produced by homogenization. By increasing the applied homogenization pressure and the number of homogenization cycles, the HSA NPs were repeatedly subjected to high shear, which further reduced the size of the droplets to nano-sized droplets [44]. In summary, even when the mean diameter of the NPs reached an optimum, additional homogenization cycles could be used to reduce the PDI. Therefore, a further optimization of the HPH conditions is necessary during the development of HSA NP formulations. In this study, 20,000 psi was set to obtain smaller-sized NPs for further experiments.

#### 3.1.3. Effect of the Ratio of Dispersed Phase (DP) to Continuous Phase (CP)

The ratio of the DP to the CP is one of the critical factors in preparing HSA NPs during the HPH processes. Thus, different DP/CP ratios of 0.1/30, 0.375/30, and 0.8/30 were evaluated to study their impacts on MPT0B291-HSA NP formulation with HPH processes. MPT0B291-HSA NPs were obtained from the ultrasonic oscillation with a fixed amplitude of 30%, followed by HPH utilizing 20 cycles of homogenization at 20,000 psi. As shown in Figure 4, with an increasing DP/CP ratio from 0.1/30 to 0.8/3, particles sizes of HSA NPs were reduced regardless of the number of cycles. A larger particle size and wider size distribution were found at the DP/CP ratio of 0.1/30 with either 10 or 20 of homogenization cycles. This might be due to the CP containing a large amount of water resulting in faster coalescence of droplets. In other words, an emulsified state might not properly form at lower fractions of the DP. Particle sizes at DP/CP ratios of 0.375/30 and 0.8/30 were both within our requirements (<200 nm), while the PDI was slightly higher for the 0.8/30 DP/CP ratio. Large aggregates were found in samples prepared using the 0.8/30 DP/CP ratio. This suggested that the DP could not be homogeneously dispersed into the CP when the DP fraction was too high. Therefore, a DP/CP ratio of 0.375/30 was selected as an optimal parameter for forming a relatively stable dispersion system.

### 3.2. Characterization of Albumin NPs and MPT0B291-HSA NPs

After determining optimal sonication and HPH conditions, the particle size and PDI of the optimized MPT0B291-HSA NPs were measured with a Zetasizer laser particle size analyzer. In addition, the morphology of the preparation was characterized by TEM using negative staining. Table 1 shows that the prepared MPT0B291-HSA NPs had an average hydrodynamic diameter of 136.3 ± 4.51 nm with a good PDI of 0.24, suggesting homogeneous distribution (monodisperse) of the NPs size within the acceptable range of protein-based nanomaterials [45,46,47]. Additionally, a value of less than 5% for the coefficient of variation for the particle size from multiple batches (3.3%) was noted, indicating a high reproducibility of the formulation. The particle size of MPT0B291-HSA NPs increased upon lecithin addition during MPT0B291-HSA NP formation. Incorporating lecithin was useful to maintain the final particle charge, as it is substantially correlated with the isoelectric point of soybean lecithin of 6.7 [45,46]. A similar approach was also used in other studies comprising egg yolk-lecithin and albumin to coat docetaxel in NP preparations [47,48]. The negative charge helps maintain the stability of the NPs in vivo. To further define the morphology of the formulated NPs, a TEM study was conducted. TEM images (Figure 5) showed that the morphology of the optimized MPT0B291-HSA NPs were spherical and evenly distributed with no obvious aggregations. These results agreed with the DLS data. The lyophilized powder of MPT0B291-HSA NPs also displayed good redispersibility when nanosuspensions were reconstituted with deionized water.

The encapsulation efficiency (EE) and drug loading (DL) of MPT0B291-HSA NPs were 92.0 ± 1.11% and 5.97 ± 0.07%, respectively. Although there were no other NP products for comparison, these data suggest that MPT0B291 was effectively loaded into the HSA NPs. Their EE and DL values were almost equal to other anticancer drug-loaded NPs, which might fit future in vivo requirements.

Further, a Bradford assay was utilized to measure the protein recovery rate. As shown in Table 1, the recovery rate of albumin was 95.0 ± 1.24. The ratio of protein to drug recovered was 9.6 which is close to the original ratio of albumin to the drug (10:1). This result indicates no excessive loss of albumin during NP preparation.

### 3.3. Stability Studies of MPT0B291-HSA NPs

The stability of MPT0B291-HSA NPs stored at 4 °C was evaluated based on maintaining the particle size and drug content. Figure 6 shows that MPT0B291-HSA NPs remained almost intact during 4 weeks of storage (with a particle size of <200 nm). The PDI of the MPT0B291-HSA NP dispersion also remained <0.3 during the 4 weeks of storage. In terms of the drug content after reconstitution of MPT0B291-human serum albumin (HSA) nanoparticles (NPs), 3.70 ± 1.98% of the drug loss was found after 1 week of storage at 4 °C. These results emphasized that the self-assembly of albumin could emerge as an ideal template for drug payloads without an additional chemical crosslinker. This suggests that MPT0B291-HSA NPs can be expected to be stable in vitro as no apparent aggregation or drug loss was observed during the experimental period. In addition, the powder and colloidal dispersion forms of MPT0B291-HSA NPs were almost the same as those before lyophilization (data not shown), indicating that our MPT0B291-HSA NPs will be robust when applied to a pharmaceutical process.

### 3.4. In Vitro Drug Release

Drug-release profiles of 0.25 mg/mL MPT0B291 from HSA NPs and solution (propylene glycol) at 37 °C were studied in triplicate. The amount of MPT0B291 released from the NP solution at fixed intervals over a period of 48 h was determined and compared to the free MPT0B291 dissolved in propylene glycol, as shown in Figure 7. Approximately 60% of the drug was released within 48 h from MPT0B291-HSA NPs, showing that a significantly sustained release profile was found from MPT0B291-HSA NPs. Meanwhile, free MPT0B291 was rapidly released (60% within 1 h) and the complete release (nearly 100%) of the drug occurred within 8 h. The sustained-release property of MPT0B291-HSA NPs in vitro arises from the high affinity between MPT0B291 and albumin, in which the albumin further acts as a protective layer for the drug from immediate diffusion into the system, enabling the MPT0B291 to be released more stably and continuously, which might allow for the continuous targeting of cancer cells in vivo. In consequence, this would provide an effective method, devoid of side effects, which was precisely the advantage of the NP delivery system.

### 3.5. In Vitro Cytotoxicity Test

The cytotoxicity of free MPT0B291 and MPT0B291-HSA NPs was investigated by MTT assays with Mia Paca-2 cells. As shown in Figure 8a, drug-free HSA NPs exhibited only a negligible cytotoxic effect, suggesting their potential suitability as drug carriers [49]. In Figure 8b and Appendix A, survival rates of Mia Paca-2 cells in different concentrations of free MPT0B291 were similar to those in MPT0B291-HSA NPs. Further, the values of the 50% inhibitory concentration (IC_50_) of free MPT0B291 and MPT0B291-HSA NPs after 24 h of exposure were determined as 4.71 and 4.28 µM, respectively (Appendix A). These results indicated that MPT0B291-HSA NPs indeed had antiproliferative activity against Mia Paca-2 cells. The nearly identical cytotoxic activities exhibited by MPT0B291-HSA NPs and MPT0B291 at equal concentrations indicated that the use of HSA-NPs as a drug vehicle did not affect the anticancer effect of MPT0B291, while maintaining similar anticancer activities in vitro but with lower cytotoxicity.

### 3.6. Maximum Tolerated Dose (MTD)

Mice were treated with MPT0B291-HSA NPs and free MPT0B291 to establish the MTD. According to the results (Figure 9), none of the mice that received injections experienced significant weight loss. Studies showed that after a single i.v. dose of 150 mg/kg BW of the MPT0B291 solution, one mouse was found dead after 24 h and another one also died on the following day. Therefore, the MTD of MPT0B291 solution administered was approximately 75 mg/kg BW by the i.v. route. On the other hand, we did not establish the MTD of MPT0B291-HSA NP formulations administered in a single dose due to the limitation to volume via i.v. administration in mice. The mice showed no painful reactions, and no mice died even after the administration of 200 mg/kg of MPT0B291-HSA NPs during 16 days of observation. Therefore, the MTD of MPT0B291-HSA NP formulations was >200 mg/kg BW (single dose). This demonstrated that mice could tolerate a higher concentration of encapsulated MPT0B291 than of free MPT0B291.

### 3.7. In Vivo PK Study

MPT0B291 concentration-time curves after intravenous administration of different formulations in rats are shown in Figure 10. The PK parameters were calculated and are shown in Table 2. Concentration-time curves of MPT0B291-HSA NPs differed significantly from those of the MPT0B291 solution. The MPT0B291-HSA NPs exhibited much higher AUC0~24h (*p* < 0.01) (2225.09 ± 767.70 vs. 290.98 ± 39.73 h·ng/mL) and slower clearance (*p* < 0.0001) (38.24 ± 10.53 vs. 288.47 ± 36.66 mL/min/kg) as compared to the free MPT0B291 (Table 2). This demonstrates an increased systemic circulation time and enhanced MPT0B291 bioavailability in encapsulated form. A much longer MRT (4.95 ± 1.03 h) was also found after i.v. administration of MPT0B291-HSA NPs, which indicates a prolonged drug duration.

The above results were in agreement with those previously obtained for in vitro drug–release curves, that coating the MPT0B291 with albumin NPs protected the drug from quick diffusion to the system in vivo, yet exhibited sustained release of MPT0B291 (Figure 7). In contrast, the free MPT0B921 induced burst release once injected into the blood which eventually initiated opsonization by the mononuclear phagocyte system (MPS), thus lowering its plasma concentration progressively during the time course (Figure 10). The use of the albumin NPs incorporated in this study seemed to successfully cloak the MPT0B291 from unselective uptake by the MPS, enabling a prolonged circulation time effective drug distribution in the blood and slower elimination [26,50,51]. Furthermore, it was evident that the blood circulation half-life (*t*_1/2_) of the encapsulated drug was 2.5 times higher than that of the free drug (Table 2), corresponding to the sustained release profile of the MPT0B291 HSA NPs including the influence of absorption and distribution.

PK parameters showed that MPT0B291-HSA NPs had the advantages of maintaining higher and steadier plasma concentrations than free MPT0B291, which indicated that MPT0B291-HSA NPs could more stably and continuously release MPT0B291. MPT0B291-HSA NPs are likely to have significant potential for in vivo antitumor efficacy in the future.

## 4. Conclusions

NPs based on albumin have received considerable interest due to their high binding capacity to hydrophobic drugs and biocompatibility with no serious side effects. In this study, HSA was selected as an ideal candidate for drug delivery due to its ready availability, biodegradability, and priority uptake [27,52]. We demonstrated that MPT0B291, a water-insoluble anticancer drug, could be successfully loaded into albumin NPs via a two-stage emulsification method under optimized conditions. These MPT0B291-HSA NPs had a favorable particle size of <200 nm with a high encapsulation efficiency and drug loading, which are desirable for intravenous injections. In vitro studies revealed that the MPT0B291-HSA NPs exhibited sustained release of MPT0B291, which suggested that MPT0B291-HSA NPs are likely to prolong MPT0B291 in the circulation in vivo. Furthermore, in the in vivo experiments, we showed that incorporating MPT0B291 in albumin NPs allowed us to prepare single-dose administration with a higher tolerated dose, resulting in increased biological safety and reduced adverse effects. The PK study also clearly demonstrated that the optimized HSA NPs significantly enhanced the drug bioavailability of MPT0B291. This suggests that MPT0B291-HSA NPs offer promising antitumor activity with low systemic toxicity. In summary, we successfully prepared an injectable, biocompatible, biodegradable, targeted, safe, effective, and controlled-release nanoscale preparation, which may be significant in promoting more nanoscale preparations for hydrophobic drugs to be applied in clinical practice.

## Figures and Tables

**Figure 1 pharmaceutics-13-01728-f001:**
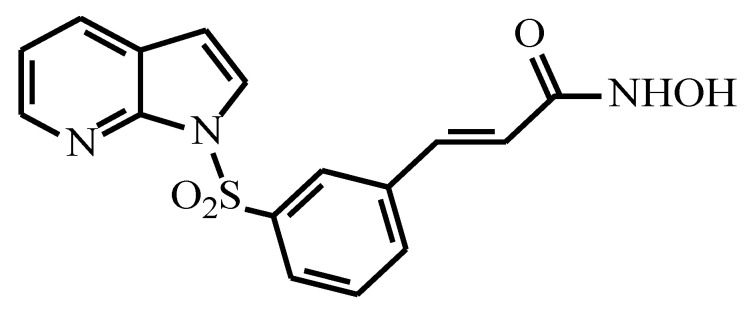
Chemical structure of MPT0B291, with molecular weight 343.36 g/mol.

**Figure 2 pharmaceutics-13-01728-f002:**
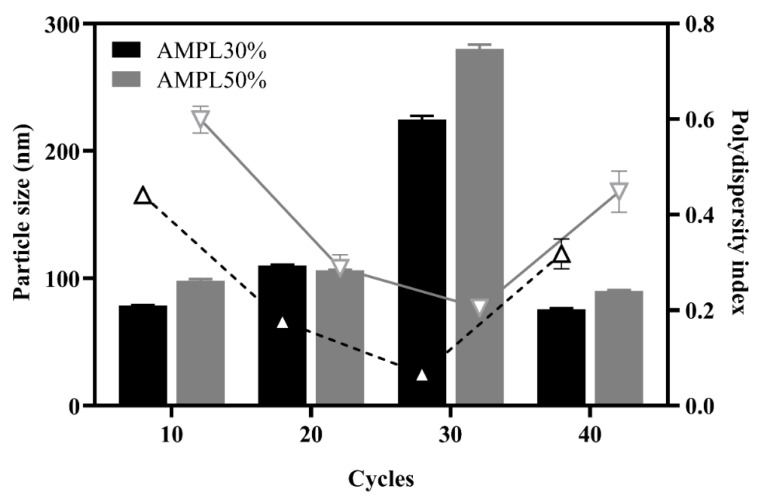
Effects of different amplitudes (AMPLs) of ultrasonication and different cycles of high-pressure homogenization on the particle size and size distribution. Column: particle size (nm); line: polydispersity index. Each point represents the mean ± SD. Error bars represent the standard deviation (*n* = 3).

**Figure 3 pharmaceutics-13-01728-f003:**
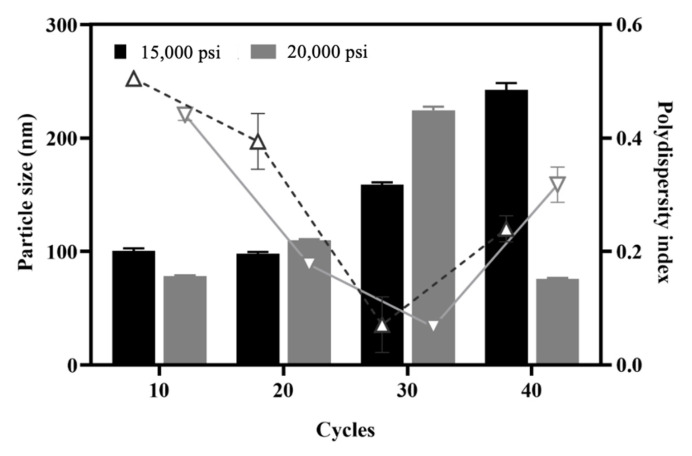
Effects of different pressures of high-pressure homogenization at different cycles on the particle size and size distribution. Column: particle size (nm); line: polydispersity index. Each point represents the mean ± SD. Error bars represent the standard deviation (*n* = 3).

**Figure 4 pharmaceutics-13-01728-f004:**
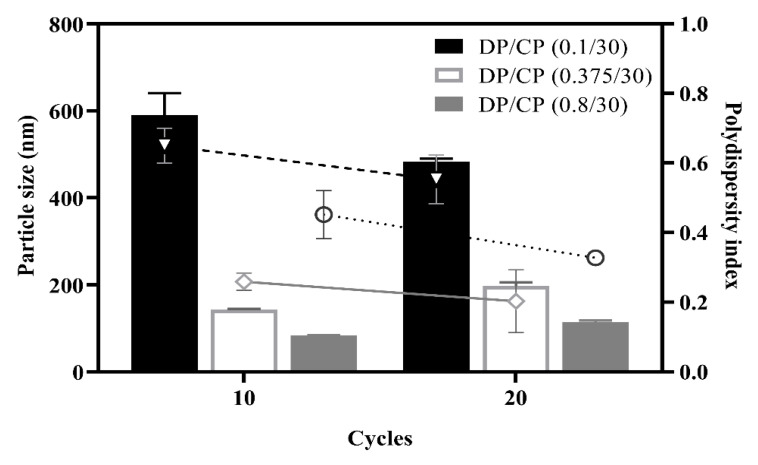
Effects of different ratios of the dispersed phase (DP) and continuous phase (CP) and different cycles of high-pressure homogenization on the particle size and size distribution. Column: particle size (nm); line: polydispersity index. Each point represents the mean ± SD. Error bars represent the standard deviation (*n* = 3).

**Figure 5 pharmaceutics-13-01728-f005:**
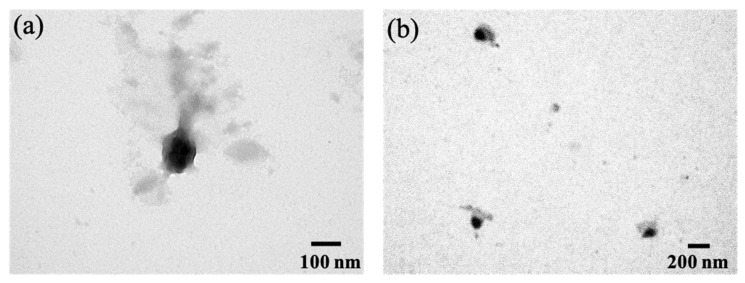
Transmission electron microscopic (TEM) images of MPT0B291-human serum albumin (HSA) nanoparticles (NPs). Scale bars represent 100 (**a**) and 200 nm (**b**).

**Figure 6 pharmaceutics-13-01728-f006:**
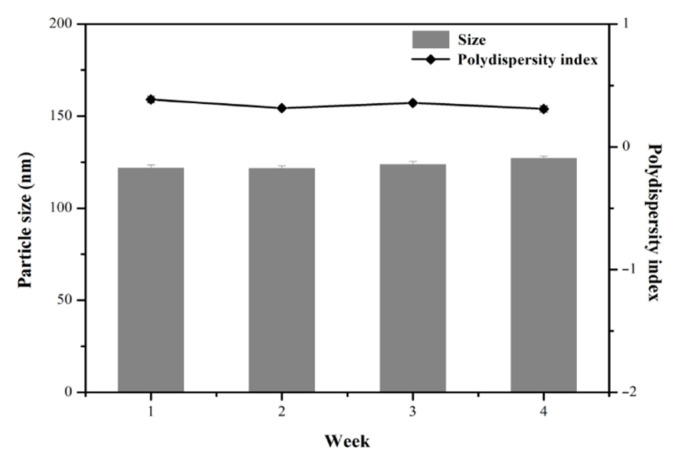
Particle size and size distribution of MPT0B291-human serum albumin (HSA) nanoparticles (NPs) stored at 4 °C for 4 weeks after reconstitution. Each point represents the mean ± SD. Error bars represent the standard deviation (*n* = 3).

**Figure 7 pharmaceutics-13-01728-f007:**
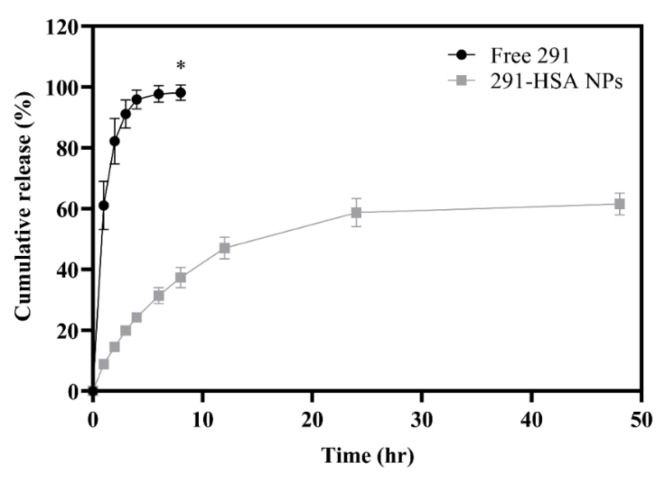
In vitro drug-release profiles of free MPT0B291 and MPT0B291-human serum albumin (HSA) nanoparticles (NPs) in pH 7.4 phosphate buffer containing 1% Tween 80 at 37 °C. Data shown are the mean ± SD (*n* = 3). * *p* < 0.0001 when free MPT0B291 was compared to MPT0B291-HSA NPs at 8 h.

**Figure 8 pharmaceutics-13-01728-f008:**
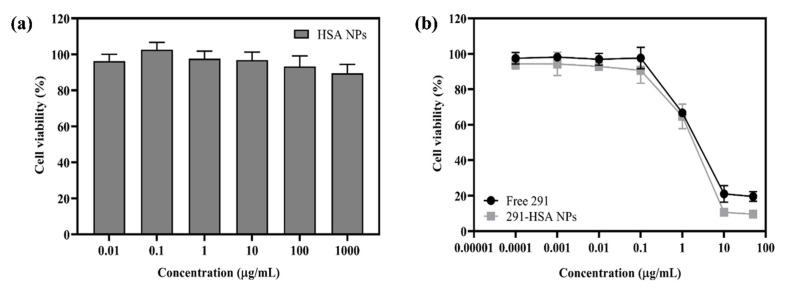
Cell viability of blank human serum albumin (HSA) nanoparticles (NPs) in Mia Paca-2 cells (**a**) and comparison of the cell viability of free MPT0B291 and MPT0B291-HSA NPs in Mia Paca-2 cells (**b**). Each point represents the mean ± SD. Error bars represent the standard deviation (*n* = 3).

**Figure 9 pharmaceutics-13-01728-f009:**
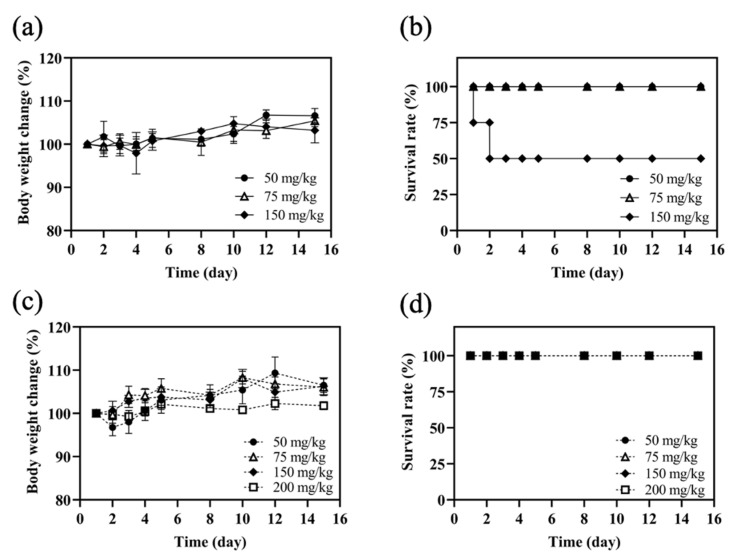
Determination of the maximum tolerated dose (MTD) of MPT0B291 and MPT0B291-human serum albumin (HSA) nanoparticles (NPs) in mice. Body weight changes and survival rates of female BALB/c mice after single-dose intravenous administration with 50, 75 and 150 mg/kg of MPT0B291 solution (**a**,**b**) and with 50, 75, 150 and 200 mg/kg of MPT0B291-HSA NPs (**c**,**d**), respectively. Each point represents the mean ± SD. Error bars represent the standard deviation (*n* = 4).

**Figure 10 pharmaceutics-13-01728-f010:**
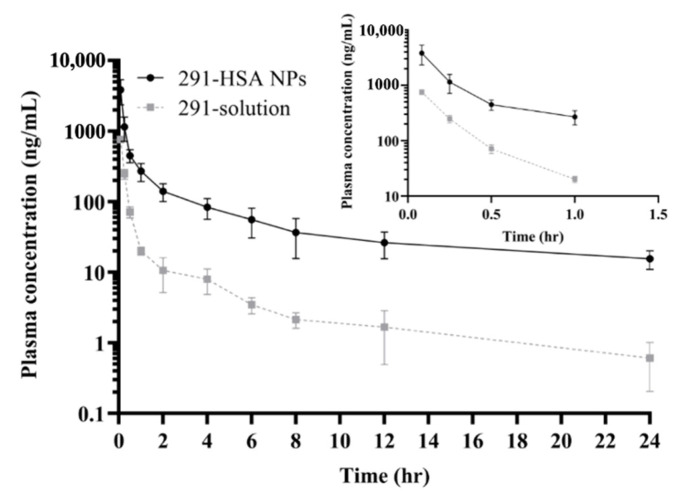
Plasma concentration-time curves of MPT0B291 after intravenous administration with MPT0B291-human serum albumin (HSA) nanoparticles (NPs) and an MPT0B291 solution (at a single dose of 5 mg/kg body weight) to rats. Each point represents the mean ± SD of three determinations (*n* = 4).

**Table 1 pharmaceutics-13-01728-t001:** Characterization of MPT0B291-loaded human serum albumin (HSA) nanoparticles (NPs). Each point represents the mean ± SD. Error bars represent the standard deviation (*n* = 3).

	Particle Size (nm)	PDI	Zeta Potential (mV)	EE (%)	DL (%)	Albumin Recovery (%)	Ratio of Albumin to Drug
**MPT0B291-HSA NPs**	136.3 ± 4.51	0.241 ± 0.077	−1.87 ± 0.133	92.0 ± 1.11	5.97 ± 0.07	95.0 ± 1.24	9.6

Abbreviations: PDI, polydispersity index; EE, encapsulation efficiency; DL, drug loading.

**Table 2 pharmaceutics-13-01728-t002:** Primary pharmacokinetic parameters of MPT0B291 after intravenous administration with MPT0B291-human serum albumin (HSA) nanoparticles (NPs) (at 5 mg/kg body weight) and an MPT0B291 solution (5 mg/kg body weight) to Sprague-Dawley rats. Data shown are the mean ± SD (*n* = 4).

Parameter	MPT0B291-HSA NPs	MPT0B291 Solution
AUC_0–24h_ (h⋅ng/mL)	2225.09 ± 767.70 **	290.98 ± 39.73
AUC_0–∞_ (h⋅ng/mL)	2333.73 ± 764.32 **	292.60 ± 39.17
*t*_1/2_ (h)	4.71 ± 0.65 **	2.07 ± 0.64
CL (mL/min/kg)	38.24 ± 10.53 ***	288.47 ± 36.66
MRT (h)	4.95 ± 1.03 **	1.86 ± 0.66
V_ss_ (L/kg)	11.49 ± 3.92	32.58 ± 12.26 *

Abbreviations: AUC_0–24h_, area under the plasma concentration-time curve (AUC) from time 0 to 24 h; AUC_0–∞_, AUC extrapolated to infinity; CL, clearance; *t*_1/2_, elimination half-life; V_ss_, volume of distribution at steady state; MRT, mean residence time. ** *p* < 0.01, ** *p* < 0.001, *** *p* < 0.0001.

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
