# Peer review of "Optimization and Development of Selective Histone Deacetylase Inhibitor (MPT0B291)-Loaded Albumin Nanoparticles for Anticancer Therapy"

_pharmaceutics, 2021, doi:10.3390/pharmaceutics13101728_

Round 1

Reviewer 1 Report

The major concern about reproducibility has not been adequately addressed. Multiple formulations need to analyzed to show reproducibility. If this was done, it is not adequately explained in the manuscript. Will error bars are included on figures, what these actually represent were not described. And there is no discussion about variability between formulations. 

The following comments from the original review still need to be addressed. 

The data in Figure 10 are somewhat surprising in that the curves have the same temporal trend, just lower release amount from the encapsulated sample. How is this explained, especially given Figure 7? The discussion on this figure needs to be carefully considered, and perhaps a longer time frame needs to be analyzed. One consideration is to normalize the starting concentrations.

Control experiments need to be carefully considered. All data sets should include inhibitor, nanoparticles and encapsulated inhibitor.

Reviewer 2 Report

In this study, the preparation procedure of human serum albumin NPs intended for the loading of MPT0B291 was developed and optimized. The formulation was optimized based on the particle size of the NPs and the encapsulation efficiency of the payload. Then the optimal formulation was characterized by TEM, and its storage stability was assessed. Finally, the in vitro release of the payload and the cell viability study were carried out. In vivo tests were performed to evaluate the PK and the maximum tolerated dose. The principles for formulation optimization were not described clearly, and the results were not discussed in detail.

Some major concerns are listed below.

1- Regarding the freeze dry process, trehalose concentration and the NP-to-trehalose ratio used should be added in method section. Was the freeze dry process optimized in this study or in a previous one? which is the rational of using trehalose and its selected concentration.

2- The preparation method of MPT0B291-loaded albumin NPs do not include a purification phase to remove the unloaded drugs. Therefore, the non-encapsulated MPT0B291 as well as free albumin are in the final sample. A possible purification should be at least mentioned in the manuscript.

3- The selected NPs preparation conditions (sonication, homogenization, DP/CP) should be clearly stated in the text. Moreover, a quality-by-design approach (Design of Experiment) should be applied to point out the effect of the parameters on the NPs features with more scheduled experiments. Figure 1 shows the effect of effects of different amplitudes of ultrasonication and different cycles of homogenization (20000 psi) on the particle size and size distribution. Information about homogenization at 15 000 psi are lack. In this case which is the CP/DP used. Figure 2 shows the effects of different pressures of high-pressure homogenization at different cycles, what about amplitudes of ultrasonication and CP/DP were selected. Considering the aim of the work, these results should be better represented and discussed.

4- Line 369. The no excessive loss of albumin during NP preparation could be also related to the lack of purification. Are the authors sure that all the albumin present is in form of NPs?

5- For the stability study, particles size was monitored after 1,2,3,4 weeks while drug content was evaluated only after 7 days. Using the same timepoints should be appropriated.

6- Which is the rational of using MIA PaCa-2 cells.

7- More representative TEM should be added, the number of NPs is too low to appreciate the size uniformity of the sample. Increasing the NPs concentration could be useful.

8- Line 138 “positively charged dye, uranyl acetate”. Uranyl acetate is not a positive charged dye, it is a salt.

Round 2

Reviewer 1 Report

Comments have been addressed sufficiently. 

Reviewer 2 Report

The manuscript has been improved and it is suitable for publication